# Plasma Membrane Transporters as Biomarkers and Molecular Targets in Cholangiocarcinoma

**DOI:** 10.3390/cells9020498

**Published:** 2020-02-21

**Authors:** Jose J.G. Marin, Rocio I.R. Macias, Candela Cives-Losada, Ana Peleteiro-Vigil, Elisa Herraez, Elisa Lozano

**Affiliations:** HEVEFARM Group, Center for the Study of Liver and Gastrointestinal Diseases (CIBERehd), Carlos III National Institute of Health. University of Salamanca, IBSAL, 37007-Salamanca, Spaincandelacives@usal.es (C.C.-L.); anapeleteiro@usal.es (A.P.-V.); elisah@usal.es (E.H.); elisa_biologia@usal.es (E.L.)

**Keywords:** ABC proteins, biliary cancer, chemoresistance, chemosensitivity, chemotherapy, hepatocellular carcinoma, SLC proteins, transporter, uptake

## Abstract

The dismal prognosis of patients with advanced cholangiocarcinoma (CCA) is due, in part, to the extreme resistance of this type of liver cancer to available chemotherapeutic agents. Among the complex mechanisms accounting for CCA chemoresistance are those involving the impairment of drug uptake, which mainly occurs through transporters of the superfamily of solute carrier (SLC) proteins, and the active export of drugs from cancer cells, mainly through members of families B, C and G of ATP-binding cassette (ABC) proteins. Both mechanisms result in decreased amounts of active drugs able to reach their intracellular targets. Therefore, the “cancer transportome”, defined as the set of transporters expressed at a given moment in the tumor, is an essential element for defining the multidrug resistance (MDR) phenotype of cancer cells. For this reason, during the last two decades, plasma membrane transporters have been envisaged as targets for the development of strategies aimed at sensitizing cancer cells to chemotherapy, either by increasing the uptake or reducing the export of antitumor agents by modulating the expression/function of SLC and ABC proteins, respectively. Moreover, since some elements of the transportome are differentially expressed in CCA, their usefulness as biomarkers with diagnostic and prognostic purposes in CCA patients has been evaluated.

## 1. Introduction

The two most frequent liver cancers, i.e., hepatocellular carcinoma (HCC), followed in the order of incidence by cholangiocarcinoma (CCA), are extremely resistant to pharmacological treatment [1]. The mechanism of action of drugs used to treat these patients involves the interaction with their molecular targets, which, in most cases, are located intracellularly and, hence, require their entrance across the plasma membrane. Since, in aqueous solution, most of these drugs are electrically charged and, therefore, are not able to freely diffuse across the lipid bilayer, they must enter cancer cells through plasma membrane transporters. These proteins belong to the solute carrier (SLC) superfamily of proteins, which, in humans, is formed by more than 400 members organized into 65 families. They share a structure characterized by a variable number of transmembrane domains connected by intracellular and extracellular loops. On the other hand, several efflux pumps are involved in lowering the intracellular concentration of active agents by favoring the exit of the drugs. Most of these proteins are primary transporters driven by the energy generated during ATP hydrolysis and belong to the superfamily of ATP-binding cassette (ABC) proteins. The general structure of these integral membrane proteins is characterized by two halves, each one formed by a region composed of several hydrophobic transmembrane alpha-helices bound to a bulky intracellular nucleotide-binding domain (NBD), where the catalytic site is located, and whose structure is used for their classification. In humans, there are seven families of ABC proteins (from A to G), of which only families ABCB, ABCC and ABCG contain members playing recognized roles in cancer chemoresistance. As a matter of fact, one of the most common mechanisms accounting for chemotherapy resistance in cancer cells is the upregulation of these ABC pumps that actively export a large number of different antitumor drugs, contributing to the multidrug resistance (MDR) phenotype of many types of tumors, including CCA [1], reducing drug bioavailability [2]. Therefore, “cancer transportome”, defined as the set of transporters expressed at a given moment in cancer cells, is an essential element in an overall complex picture that determines the unsatisfactory response of CCA to the available chemotherapy. 

Since some transporters are differentially expressed in CCA and HCC, some attempts have been made to use them as biomarkers in diagnosis and prognosis and, even when they are expressed in both tumoral cholangiocytes and hepatocytes, they can be useful as targets to develop strategies aimed at sensitizing tumor cells to chemotherapy, either by increasing the uptake or reducing the export of anticancer drugs.

## 2. Carriers as Biomarkers

The usefulness of transport proteins as biomarkers for the diagnosis, prognosis, or monitoring the response to therapy for several types of cancer, including primary liver tumors, has been described in recent studies. Transport proteins that are expressed at the plasma membrane of cancer cells are identical or similar to those found in healthy cells under physiological conditions; although, in tumor cells, changes in their expression levels and the appearance of genetic and epigenetic variations are frequent and can contribute to chemotherapy resistance. Some transport proteins are ubiquitous, whereas others are tissue-specific or even are expressed only in one type of cells in a specific tissue. The latter ones are of particular interest as biomarkers, since they may permit to distinguish different types of tumors based on the organ or the cell type from where they are originated (Table 1).

Regarding the liver, transport proteins involved in the uptake and efflux of bile acids [3] meet these requirements and have been proposed as biomarkers [4]. This is the case of the apical sodium-dependent bile acid transporter (ASBT, gene symbol *SLC10A2*), located in the apical membrane of cholangiocytes, whose expression is maintained or slightly increased in CCA [5]. Since ASBT is not expressed in hepatocytes, the detection of this transporter can be useful to differentiate CCA from HCC. 

On the other hand, several members of the family of organic anion-transporting polypeptides (OATPs) are expressed in healthy hepatocytes and cholangiocytes. Namely, OATP1B1 and OATP1B3 (gene symbols *SLCO1B1* and *SLCO1B3*, respectively), which play an important role in the uptake of bile acids and many other endogenous and xenobiotic compounds by the hepatocytes, are expressed in HCC [6,7]. However, understanding the relevance of members of the OATP family as biomarkers in CCA requires further investigation.

The copper transporter 1 (CTR1, gene symbol *SLC31A1*) is able to transport platinum derivatives, and the reduction in CTR1 expression observed in CCA [7] might result in enhanced chemoresistance to these drugs. Also, a relationship between the existence of polymorphisms in the *SLC31A1* gene and the response to gemcitabine–platinum treatment in biliary tract cancer patients has been reported [8].

Another interesting marker for the prediction of response to treatment based on nucleoside analogs is the presence in the tumor cell plasma membrane of the equilibrative nucleoside transporter 1 (ENT1, gene symbol *SLC29A1*), which has been associated with better outcome in patients receiving gemcitabine as an adjuvant in resected CCA and as first-line treatment in advanced biliary tract cancer [9,10]. 

The organic cation transporter 1 (OCT1, gene symbol *SLC22A1*) is present both in healthy cholangiocytes and hepatocytes. The detection of OCT1 in the plasma membrane of cancer cells may be considered as a prognostic biomarker in sorafenib-based HCC therapy [11,12]. Low OCT1 expression in CCA has also been associated with the refractoriness of this tumor to sorafenib in animal models and in in vitro assays [13]. Moreover, the analysis of OCT1 expression in different types of liver lesions, together with that of other SLC22A family members, such as OCT3 (gene symbol *SLC22A3*), has been proposed as a diagnostic marker of intrahepatic lesions [14]. However, a recent study provides evidence for age-related changes in transcriptome liver profile for important drug-related genes, such as *SLC22A1*, whose expression was higher in the group of older people [15], thus requiring stratification for the use of these genes as biomarkers.

High levels of glucose transporter 1 (GLUT1, gene symbol *SLC2A1*) have been found in several cancers, including intrahepatic CCA (iCCA), associated with hypoxia, which is consistent with the elevation of anaerobic metabolism together with the poor cell differentiation and metastasis that are characteristic of these tumors [16]. GLUT1 has been proposed as a biomarker for the detection of bile duct carcinoma and for distinguishing between CCA from HCC [17]. In addition, the expression of GLUT2 (gene symbol *SLC2A2*) was proposed as a marker of high-grade biliary intraepithelial neoplasia lesions [16]. 

The expression levels of monocarboxylate transporter 4 (MCT4, gene symbol *SLC16A3*) increase with HCC progression, while that of MCT2 (gene symbol *SLC16A7*) is lost in advanced HCC and is associated with a better prognosis [18]. The value of this marker in CCA is not known. 

The overexpression of the anion exchanger 2 (AE2, gene symbol *SLC4A2*) has been associated with HCC development [19], while inadequate AE2 expression in cholangiocytes may lead to the development of chronic cholangiopathies [20], which are risk factors of CCA. 

Aquaporin-1 (AQP-1, gene symbol *AQP1*) has been proposed as a selective marker for differentiated cholangiocytes and as a useful biomarker to immunohistochemically differentiate between CCA and HCC and metastatic colorectal carcinomas [21], while high AQP-5 (gene symbol *AQP5*) expression was associated with better prognosis and drug sensitivity in biliary tract carcinoma [22]. 

Phosphohippolin (PPH, *FXYD6* gene), a transmembrane protein that functions as an ion channel and affects the activity of Na^+^/K^+^-ATPase, has been found upregulated in CCA compared with normal bile duct tissue, and has been proposed as a new potential biomarker and therapeutic target for CCA [23] (Figure 1).

Among members of the family ABCB of the ABC proteins, the high expression of P-glycoprotein or multidrug resistance protein 1 (MDR1, gene symbol *ABCB1*) detected by immunohistochemistry in HCC was associated with a bad prognosis [24,25]; moreover, this pump has been proposed as a valuable biomarker of prognosis in gallbladder cancer [26]. In the same family, the bile salt export pump (BSEP, gene symbol *ABCB11*), which is also located at the canalicular membrane of healthy hepatocytes, has been suggested to be a sensitive and specific marker of HCC [27], whereas the phospholipid translocase MDR3 (gene symbol *ABCB4*), which is located in the apical membrane of hepatocytes and cholangiocytes, may be a useful marker to discriminate immunohistochemistry, HCC and iCCA from hepatoid carcinomas [28]. Mutations in both pumps (BSEP and MDR3) have been described in some cases of pediatric patients with CCA [29,30], which may lead to increased CCA susceptibility. 

Contradictory information has been published regarding the expression in CCA of another ABC protein, the multidrug resistance-associated protein 2, (MRP2, gene symbol *ABCC2*), which is located at the apical pole of the hepatocyte plasma membrane. On the one hand, MRP2 has been detected by immunohistochemistry, not only in HCC (72/80 cases), but also in CCA (52/54 cases) [31], whereas another study has reported the presence of MRP2 in only a low proportion of gallbladder tumors (4/14 cases), but an undetectable expression in all CCA assayed (0/7 cases) [2]. MRP3 (gene symbol *ABCC3*), located at the basolateral membrane of both hepatocytes and cholangiocytes, has been found moderately expressed in only some cases of HCC (15/80 cases) and highly expressed in CCA specimens (24/54 cases) [31]. These results do not support the use of these ABC proteins as markers for the differential diagnosis between HCC and CCA; however, the positivity of the staining could be useful for an association with other clinical parameters. In fact, the over-expression of MRP3 has been proposed as a marker of resistance to sorafenib in HCC-derived cell lines [32]. Whether this concept can be extrapolated to patients has not been confirmed. Regarding iCCA, the determination of mRNA levels by RT-qPCR revealed that only MRP1 (gene symbol *ABCC1*) was a candidate prognostic biomarker; while the loss of breast cancer resistance protein (BCRP, gene symbol *ABCG2*) expression by immunohistochemistry, but not of MDR1 or MRP1, was associated with more aggressive tumor progression [33] (Figure 1).

## 3. The Usefulness of Transporters in Drug Targeting

Owing to the fact that the uptake of most anticancer drugs is a crucial step for their efficacy and this mainly occurs through plasma membrane proteins, it has been hypothesized that some of these transporters could be used as targets for directing chemotherapy (Figure 1). As mentioned above, the expression levels of some of these transporters are maintained or even increased in CCA, which might facilitate the intracellular accumulation and hence the action of their substrate drugs (Table 2). Thus, gemcitabine can be taken up by nucleoside transporters, such as CNTs and ENTs (*SLC28* and *SLC29* families) [34], whose expression levels have been associated with gemcitabine response in CCA patients [35,36].

Since bile acid transporters are expressed at the plasma membrane of cholangiocytes and this feature is preserved in CCA, the possibility to vectorize anticancer drugs conjugated with bile acids towards these tumors has been explored [5]. Several cytostatic bile acid derivatives named “Bamets”—from bile acid (BA) and metal (MET)—have been previously synthesized and evaluated in liver and intestinal cells. The results revealed an efficient uptake by bile acid transporters expressed in the cells of the enterohepatic circuit [37,38]. One of these compounds, named Bamet-UD2, obtained by conjugation of cisplatin and two ursodeoxycholate moieties, has proved to be useful in experimental models of CCA. Thus, Bamet-UD2 has potent in vitro cytostatic activity and in vivo antitumor effect, and markedly lower side effects than the parent drug cisplatin [5,39]. Interestingly, Bamet-UD2 uptake by CCA cells was mediated by ASBT, whose expression levels are relatively well preserved in CCA [5].

The L-type amino acid transporter-1 (LAT1, gene symbol *SLC7A5*), is a sodium-independent carrier expressed at the plasma membrane of many cells, where this is involved in the transport of neutral amino acids, including several essential ones. LAT1 is highly expressed in many types of cancer, including CCA, where it may play a role in carcinogenesis and progression [40]. Using in vitro assays, experimental downregulation of LAT1 has been demonstrated to suppress CCA cell invasion and migration, which suggests that LAT1 constitutes a potential therapeutic target for treating CCA [41]. Furthermore, JPH203 (also known as KYT0353), a selective inhibitor of this transporter, has been shown to induce apoptosis, cell cycle arrest and impaired tumor growth in both in vitro and in vivo models of CCA [42].

Through a completely different approach, although also involving plasma membrane transporters, several studies have aimed at inhibiting tumor growth by manipulating cancer cell metabolism, for instance, by impairing glucose uptake through GLUT1, whose expression in CCA is high, as mentioned above [17]. In vitro experiments performed with CCA cells (TFK-1) revealed that the antioxidant flavonoid quercetin inhibited GLUT1 along with the metabolic activity, which was accompanied by DNA damage and enhanced cell death by activation of apoptosis [43]. 

Surprisingly, *L*-ascorbic acid has been reported to induce cytotoxicity in CCA cells by generating intracellular reactive oxygen species (ROS), and subsequently DNA damage, ATP depletion, and inhibition of the “mammalian target of rapamycin” (mTOR) survival pathway, which results in a synergistic effect with cisplatin both in vitro and in vivo. These effects depend on the expression of the sodium-dependent vitamin C transporter 2 (SVCT2, gene symbol *SLC23A2*). Accordingly, the knockdown of SVCT2 made CCA cells resistant to the treatment with L-ascorbic acid. These results suggest that SVCT2 expression levels may serve as a positive outcome predictor for treatment with vitamin C in CCA patients [44].

Another transporter that has been detected at high levels in the plasma membrane of a significant proportion of human CCA is the sodium-iodide symporter (NIS, gene symbol *SLC5A5*). NIS plays an essential role in the uptake of ^131^I and has been proposed as a potential target for radioiodine therapy in CCA [45].

## 4. Overcoming Chemoresistance by Manipulation of Uptake Transporters

The transporter of organic cation, OCT1, has been associated with the uptake of the tyrosine kinase inhibitor (TKI) sorafenib by cancer cells [46]; however, in CCA, a marked reduction in OCT1 function occurs due to a decrease in mRNA levels and the predominance of aberrant splicing variants [7,46]. This fact has also been observed in HCC [47], where this was associated with a reduced amount of OCT1 in the plasma membrane and unsatisfactory response to sorafenib [12]. In order to overcome this situation, a novel gene therapy approach has been developed to enhance the levels of this transporter and to improve sorafenib efficacy. The strategy consisted of adenoviral vectors bearing the OCT1 coding sequence under the transcriptional control of the oncogene *BIRC5* promoter, which is selectively active in CCA cells. The evaluation of these vectors, in both in vitro and in vivo CCA models, revealed that the transduction with these adenoviral vectors resulted in an increased expression of OCT1, which selectively occurred in the plasma membrane of CCA cells, but not in the healthy liver tissue. As a consequence, the uptake of sorafenib by the tumor was enhanced and the response to treatment with this drug resulted in efficient tumor growth inhibition [13].

Similar to what happens with OCT1, other SLC transporters are also downregulated in CCA, which affects the efficacy of the drugs that are taken up through them. Moreover, chemotherapy can alter the expression of several transporters not necessarily involved in the uptake of the administered drug. Thus, cisplatin treatment can temporarily induce the expression of certain transporters such as CNT1 (gene symbol *SLC28A1*), CNT3 (gene symbol *SLC28A3*), ENT1 and OCT1. Taking advantage of these changes, it has been proposed to carry out sequential cycles of chemotherapy in which cisplatin would be alternated with drugs taken up by these transporters, such as gemcitabine and sorafenib, to improve the overall response [48].

## 5. Strategies to Reduce Drug Efflux

Concomitant inhibition of drug efflux pumps through the use of antitumoral drugs represents a promising strategy for overcoming CCA chemoresistance. In this sense, a high number of new molecules, called chemosensitizers, have been developed to inhibit or modulate ABC transporters, restore drug accumulation and, hence, their chemotherapeutic efficacy. This has led to the development of three generations of ABC inhibitors so far. Most of them are structure-based inhibitors of MDR1, which is considered a prototypic ABC drug pump able to export (out of tumor cells) a large variety of drugs, such as doxorubicin, etoposide, paclitaxel, vinblastine and sorafenib [49]. Moreover, MDR1 has been found overexpressed both in CCA cell lines and in clinical samples of biliary tumors [50]. The first generation of chemosensitizers included Food and Drug Administration-approved drugs non-specifically designed for this purpose, such as verapamil, cyclosporine A or quinine. They had a potent in vitro activity, but unacceptable side effects in healthy tissues in vivo [51]. The second generation of these inhibitors, which included, for instance, valspodar and PSC-833, were developed to specifically target drug efflux transporters. They showed less toxicity, but also had off-target effects and lacked significant efficacy in clinical trials [52,53]. Accordingly, a third generation of ABC modulators was developed to improve both potency and specificity. Although in preclinical studies with acute myeloid leukemia patients some of these drugs, such as tariquidar, elacridar, zosuquidar and laniquidar, showed a strong ability to inhibit MDR1 at very low concentrations, they have not demonstrated sufficient clinical efficacy to be considered further [54] and, unfortunately, none of these compounds have been incorporated into clinically accepted treatments. Therefore, efficient and non-toxic chemosensitizers are still being sought, mainly among natural compounds. Although still under investigation, some of them have shown promising characteristics. This group includes tannic acid, a plant-derived polyphenol, which has demonstrated its ability to inhibit CCA growth both in vitro and in vivo [55]. This compound also reduces the expression of several export pumps, such as MDR1, MRP1 and MRP2, involved in CCA chemoresistance [7,50], enhancing 5-fluorouracil (5-FU), mitomycin C and gemcitabine cytotoxicity in CCA cells [56].

More recently, other natural compounds have been suggested as promising agents for CCA treatment in combination with antitumor drugs. Among them, β-escin, the most active compound of the horse chestnut seed (*Aesculus hippocastanum*), is a potent reverser of MDR1-mediated resistance, able to sensitize CCA cells to common anticancer drugs [57]. Moreover, isomorellin or forbesione, obtained from *Garcinia hanburyi*, have shown significant synergistic effects with doxorubicin in CCA cells in vitro by downregulating MRP1 [58].

In addition to natural products, there are numerous synthetic compounds that have exhibited chemosensitizing properties in CCA when combined with anticancer drugs, through modulation of ABC transporters. These include some drugs, such as metformin, which sensitizes CCA cells to both sorafenib and 5-FU by reducing MRP1 expression through modulation of the AMPK/mTOR pathway [59] and simvastatin, that decreases CCA cell viability and ABCA1 and ABCG1 expression by acting through a pathway independent on the serine-threonine protein kinase AKT [60]. Tamoxifen may also reverse the MDR phenotype of CCA cells by enhancing the chemotherapeutic effects due to its competitive inhibition of MDR1 [61]. Additionally, several TKIs have shown potential as chemosensitizers, mainly by inhibiting MDR1, MRPs and BCRP [62,63,64,65]. Many of these drugs are themselves substrates of these ABC pumps or directly inhibit their function by blocking the ATP-binding site and thus promoting drug accumulation.

The inhibition of the nuclear factor kappa B (NF-κB) by the synthetic compound dehydroxymethylepoxyquinomicin (DHMEQ) also enhances the sensitivity of CCA cells to several antitumor drugs, such as 5-FU, cisplatin and doxorubicin, presumably due to its ability to induce MDR1 and BCRP downregulation [66].

In the last years, alternative strategies for overcoming multidrug resistance by regulating ABC transporter expression have been proposed. These include approaches based on the use of antisense oligonucleotides, ribozymes, RNA interference and CRISPR/Cas9 technology. However, the literature regarding the use of gene therapy to chemosensitize CCA is scarce. Overexpression of ten-eleven translocation 1 (TET1), a methylcytosine dioxygenase that catalyzes DNA demethylation, enhances the sensitivity of CCA to gemcitabine, accompanied by a decrease in MDR1 expression [67]. In addition, overexpression of miR-199a-3p in the presence of cisplatin could decrease the proliferation rate and increase apoptosis of CCA cells, probably by regulating the expression of its target gene mTOR and reducing that of MRP1 [68]. Furthermore, inhibition of Wnt/β-catenin pathway by β-catenin siRNA also reverses the MDR phenotype of CCA chemoresistant cells by downregulating MDR1 [69].

## 6. Conclusions and Perspectives

Since cellular uptake is an essential step in the mechanisms of action of many anticancer drugs, whose activity depends on the intracellular levels of active agents, an important role of SLC and ABC proteins in determining CCA chemoresistance has been demonstrated.

It should be considered that the transportome involved in the MDR phenotype of CCA is not a steady-state feature of cancer cells but a dynamic trait that responds to pharmacological treatment in a Darwinian way of evolution among cancer cell populations, in which key features selecting more chemoresistant clones may be present before the treatment, appear during treatment or be potentiated by the treatment. This has serious consequences regarding the relapse and, hence, the clinical outcome and constitutes an important difficulty in the use of changes in expression levels or the presence of genetic variants of transporters as biomarkers.

Regarding the use of transporters as pharmacological targets, an important feature is the location of these proteins in the plasma membrane, because this is an essential requirement for their functionality. Thus, in addition to measure mRNA and protein levels, it would be necessary to confirm the correct subcellular location, e.g. using immunofluorescent staining. Moreover, it should be kept in mind that some transporters, even correctly targeted to the plasma membrane, present single nucleotide polymorphisms (SNPs) that inactivate their carrier function, as described for OCT1 [46].

Based on the advances in the field, it is evident that, in order to improve the response rates of the pharmacological treatment of CCA, in addition to more effective novel drugs, combined chemosensitizing strategies are required. With this aim, three lines of action need to be fostered: (i) to increase the expression of functional uptake transporters using gene therapy strategies, (ii) to chemically synthesize novel molecules with more selective and potent ability to inhibit export pumps or to vectorize active agents through uptake transporters and iii) to improve the selective delivery of chemotherapy to tumor cells using tools provided by advances in nanomedicine. On the other hand, the rapid increase in our knowledge regarding the biology of extracellular vesicles released by cancer cells, and the technology required for their isolation and characterization, make these nanoparticles excellent tools to indirectly explore the presence of SLC and ABC proteins in tumor cells and, hence, use them as biomarkers with diagnostic and prognostic purposes in CCA patients.

## Figures and Tables

**Figure 1 cells-09-00498-f001:**
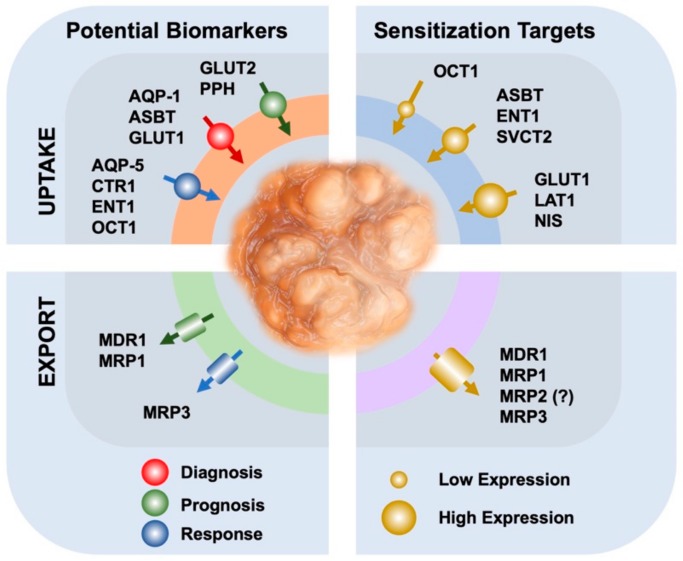
Schematic representation of the role in cholangiocarcinoma of uptake (up) and export (down) transporters as biomarkers for diagnosis and prediction of response to chemotherapy (left) or as targets for strategies of chemosensitization to antitumor drugs (right). Aquaporin-1/5 (AQP-1/5); apical sodium-dependent bile acid transporter (ASBT); cooper transporter (CTR1); equilibrative nucleoside transporter 1 (ENT1); glucose transporter 1/2 (GLUT1/2); L-type amino acid transporter-1 (LAT1); multidrug resistance protein 1 (MDR1); multidrug resistance-associated protein 1/3 (MRP1/3); sodium–iodide symporter (NIS); organic cation transporter 1 (OCT1); Phosphohippolin (PPH); sodium-dependent vitamin C transporter 2 (SVCT2).

**Table 1 cells-09-00498-t001:** Plasma membrane transporters with potential usefulness as biomarkers.

Usefulness	Gene	Protein	Levels in CCA	Levels in HCC	Potential Interest/Evidences
Diagnosis	*SLC10A2*	ASBT	Mild	N.D.	To distinguish CCA and HCC/in vitro/in vivo/IHC
	*SLC2A1*	GLUT1	High	Low	To distinguish CCA and HCC/IHC
	*AQP1*	AQP-1	High	Low	To distinguish CCA and HCC/IHC
Response to chemotherapy	*SLC29A1*	ENT1	Variable	Variable	Prediction of response to nucleoside analogues/Expression associated with gemcitabine response in patients
	*SLC22A1*	OCT1	Low	Low	Prediction of response to sorafenib/Expression and location associated with sorafenib response in patients
	*SLC31A1*	CTR1	Low	Variable	Prediction of response to Pt derivatives/Expression related with drug response
	*AQP5*	AQP-5	High	High	Prognosis and drug sensitivity to gemcitabine/IHC/Expression related with drug response
	*ABCC3*	MRP3	High	Low	Biomarker of drug resistance to sorafenib/in vitro evidences
Prognosis	*SLC2A2*	GLUT2	High	High	Marker of high-grade biliary tumors/IHC
	*FXYD6*	PPH	High	High	Biomarker for favorable outcome in CCA/IHC
	*ABCB1*	MDR1	High	High	Biomarker of bad prognosis/IHC
	*ABCC1*	MRP1	High	High	Biomarker of bad prognosis/IHC

Apical sodium-dependent bile acid transporter (ASBT); aquaporin 1/5 (AQP-1/5); cholangiocarcinoma cholangiocarcinoma (CCA); copper transporter 1 (CTR1); equilibrative nucleoside transporter 1 (ENT1); glucose transporter 1/2 (GLUT1/2); hepatocellular carcinoma (HCC); immunohistochemistry (IHC); multidrug resistance protein 1 (MDR1); multidrug resistance-associated protein 1/3 (MRP1/3); aquaporin 1/5 (AQP-1/5); not detected (N.D.). CCA and HCC levels are compared to adjacent non-tumor tissue.

**Table 2 cells-09-00498-t002:** Plasma membrane transporters with potential usefulness as targets in CCA.

Gene	Protein	Levels *^a^*	Substrates	Role	Modulation
*SLC102*	ASBT	Mild	Bile acid derivatives	Drug uptake	in vitro and in vivo evidences
*SLC22A1*	OCT1	Low	Sorafenib	Drug uptake	Decitabine and cisplatin temporarily induce its expression. In vitro and in vivo evidences
*SLC29A1*	ENT1	Variable	Nucleoside analogs	Drug uptake	Cisplatin temporarily induces its expression. Associated to gemcitabine response in patients
*SLC7A5*	LAT1	High	Neutral amino acids	Suppress CCA invasion and migration	JPH203 inhibits its expression in vitro
*SLC2A1*	GLUT1	High	Glucose	Inhibition of GLUT1 reduces tumor metabolic activity	Quercetin inhibits GLUT1 in vitro
*SLC23A2*	SVCT2	Mild	*L*-Ascorbic acid	Uptake of *L*-ascorbic acid induces cytotoxicity	in vitro and in vivo evidences
*SLC5A5*	NIS	High	^131^I	Drug uptake	in vitro and in vivo evidences
*ABCB1*	MDR1	High	Doxorubicin, etoposide, paclitaxel, vinblastine, sorafenib	Drug efflux	Expression modulated by verapamil, cyclosporine A, quinine, TKIs, and others. In vitro and in vivo evidences
*ABCC1*	MRP1	High	Mitomycin C, gemcitabine, doxorubicine, sorafenib, 5-FU	Drug efflux	Expression modulated by tannic acid, isomorellin and metformin in vitro
*ABCC2*	MRP2	Unclear	Mitomycin C, gemcitabine, 5-FU	Drug efflux	In vitro evidences. Expression modulated by tannic acid.
*ABCC3*	MRP3	High	Sorafenib	Drug efflux	In vitro evidences

*^a^* Expression levels in CCA as compared with adjacent non-tumor tissue. 5-fluorouracil (5-FU); tyrosine kinase inhibitors (TKIs); apical sodium-dependent bile acid transporter (ASBT); copper transporter 1 (CTR1), organic cation transporter 1 (OCT1); equilibrative nucleoside transporter 1 (ENT1); glucose transporter 1 (GLUT1); L-type amino acid transporter-1 (LAT1); sodium-dependent vitamin C transporter 2 (SVCT2); sodium–iodide symporter (NIS); multidrug resistance protein 1 (MDR1); multidrug resistance-associated protein 1/2/3 (MRP1/MRP2/MRP3); 5-fluorouracil (5-FU); tyrosine kinase inhibitors (TKIs).

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
