# Peer review of "Plasma Membrane Transporters as Biomarkers and Molecular Targets in Cholangiocarcinoma"

_cells, 2020, doi:10.3390/cells9020498_

Round 1

Reviewer 1 Report

In this review, Marin et al. summarizes the roles of membrane transporters and their potentials as biomarkers in CCA. This review is well written covering transporters associated with CCA. However, the field is complicated so readers can be easily confused. It would be helpful if the authors add more information for readers. For example, Figure 1 shows ASBT for uptake biomarkers and chemosensitization. Is ASBT upregulated in CCA? High ASBT expression is correlated with CCA development or progression? Why ASBT is in biomarkers but OCT1 is not? Why GLUT2 is in biomarkers but not in chemosensitization? Figure 1 is not very clear and needs to be elaborated.

It would be very useful for readers if the authors add tables to support Figure 1. For example, some transporters are upregulated in CCA, and some are downregulated. A table would be useful to show transporters, up/down in CCA, only in CCA or also in HCC, associated with resistance against drugs such as sorafenib, or functions such as increasing export or impaired import leading to impaired drug effects in cancer cells etc. For transporters as biomarkers, readers would also need more information potentially in a table showing CCA vs. control groups (healthy individuals or liver disease but not cancerous or HCC etc.), sensitivity and specificity in ROC analysis, limitations etc. For transporters as therapeutic target, it would be useful to show clearly agonists/antagonists to increase/inhibit import/export, drug names, clinical trial numbers, or associated signaling pathways etc.  

Author Response

We thank the reviewer for the positive comments. I have followed the recommendations and two new tables have been added to support information given in Figure 1. Changes in the expression in the tumor and potential usefulness as biomarkers or molecular targets have been included.

Reviewer 2 Report

The review by Marin and colleagues summarizes recent investigations of plasma membrane transporters as biomarkers and potential molecular targets of cholangiocarcinoma. The authors mainly described two families of carriers: solute carrier superfamily of proteins SLC and superfamily of ATP-binding cassette (ABC) proteins, as most appropriate candidates for biomarkers. The reduction or elevation of the carriers in serum of patients strongly correlated with the severity of the cholangiocarcinoma. This is a nice review which covers a majority of the recent reports in this field of research. There are minor suggestions for improvement.

Discussion of some additional reports regarding potential biomarkers could be included. Particularly, the authors might consider discussions of the OCT1 as a marker of sorafenib treatments of HCC (Grimm et al MBMC 2016); discussions of OCT-SLC22A as a diagnostic marker of intrahepatic lesions (Visentin et al Drug Metab Dispos 2017); and a potential use of plasma membrane transporters in the prognosis of the pediatric liver cancer (Meier et al Pharamacogenet Genomics 2018). The section 6. Conclusions and Perspective should mention some potential problems with the use of plasma membrane transporters as biomarkers and drug-targets.

Author Response

The positive evaluation of our manuscript is highly appreciated. We have improved the text by including the references suggested by the reviewer. They have been appropriately cited and commented. Moreover, the section 6 has been modified in the line recommended by the reviewer.

Round 2

Reviewer 1 Report

No further comments.